# Increased oxidative stress in elderly leprosy patients is related to age but not to bacillary load

**Pedro Henrique Lopes da Silva[1]☯, Katherine Kelda Gomes de Castro[1]☯, Mayara Abud Mendes[1], Thyago Leal Calvo[1], Júlia Monteiro Pereira Leal[1], Mariana de Andréa Vilas-Boas Hacker[1], José Augusto da Costa Nery[1], Euzenir Nunes Sarno[1], Roberto Alves Lourenço[2], Milton Ozório Moraes[1], Flávio Alves Lara[3], Danuza Esquenazi[1,4]***

**1** Laboratório de Hanseníase, Instituto Oswaldo Cruz, Fundação Oswaldo Cruz, Rio de Janeiro, Brazil, **2** Laboratorio de Envelhecimento Humano, GeronLab, Policlínica Piquet Carneiro, Universidade do Estado do Rio de Janeiro, Rio de Janeiro, Brazil, **3** Laboratório de Microbiologia Celular, Instituto Oswaldo Cruz, Fundação Oswaldo Cruz, Rio de Janeiro, Brazil, **4** Disciplina de Patologia Geral, Faculdade de Ciências Médicas, Universidade do Estado do Rio de Janeiro, Rio de Janeiro, Brazil

☯ These authors contributed equally to this work.
* danuza@ioc.fiocruz.br

**Data Availability Statement:** All relevant data are within the manuscript and its Supporting Information files.

## Abstract

### Background

Leprosy continues to be a public health problem in Brazil. Furthermore, detection rates in elderly people have increased, particularly those of multibacillary (L-Lep) patients, who are responsible for transmitting *M. leprae*. Part of the decline in physiological function during aging is due to increased oxidative damage and change in T cell subpopulations, which are critical in defense against the disease. It is not still clear how age-related changes like those related to oxidation affect elderly people with leprosy. The aim of this work was to verify whether the elderly leprosy patients have higher ROS production and how it can impact the evolution of leprosy.

### Methodology/Principal findings

87 leprosy patients, grouped according to age range and clinical form of leprosy, and 25 healthy volunteers were analyzed. Gene expression analysis of antioxidant and oxidative burst enzymes were performed in whole blood using Biomark's microfluidic-based qPCR. The same genes were evaluated in skin lesion samples by RT-qPCR. The presence of oxidative damage markers (carbonylated proteins and 4-hydroxynonenal) was analyzed by a DNPH colorimetric assay and immunofluorescence. Carbonylated protein content was significantly higher in elderly compared to young patients. One year after multidrug therapy (MDT) discharge and *M. leprae* clearance, oxidative damage increased in young L-Lep patients but not in elderly ones. Both elderly T and L-Lep patients present higher 4-HNE in cutaneous lesions than the young, mainly surrounding memory CD8[+] T cells. Furthermore, young L-Lep demonstrated greater ability to neutralize ROS compared to elderly L-Lep patients, who presented lower gene expression of antioxidant enzymes, mainly glutathione peroxidase.

**Funding:** The authors received no specific funding for this work.

**Competing interests:** The authors have declared that no competing interests exist.

## Conclusions/Significance

We conclude that elderly patients present exacerbated oxidative damage both in blood and in skin lesions and that age-related changes can be an important factor in leprosy immuno-pathogenesis. Ultimately, elderly patients could benefit from co-supplementation of antioxidants concomitant to MDT, to avoid worsening of the disease.

## Author summary

Leprosy, which has been reported throughout human history since Biblical times, still presents unclear aspects in regards to its pathogeny, and represents a public health concern, particularly in developing countries, where poor sanitary conditions of socially disfavored groups is a relevant contributing factor in maintaining disease dissemination. Although its signs and symptoms are primarily found in young adults, the increasing number of elderly individuals with leprosy is a recent phenomenon that requires clarification. Considering the specific biochemical profile of the elderly, as well as the influence thereof on their immune response to infection, this work demonstrates that elderly leprosy patients present increased levels of oxidative damage in blood and skin lesions when compared to younger patients. Such findings appear to be correlated with low gene expression of antioxidant enzymes, particularly among multibacillary patients. This work intends to contribute to a better understanding on the progress of leprosy in elderly groups.

## Introduction

Leprosy is a neglected infectious disease caused by *Mycobacterium leprae*, which mainly affects skin and peripheral nerves [1]. Although the application of multidrug therapy (MDT) in the 1980s significantly reduced disease prevalence and cured millions of patients, there is a stagnant number of new cases per year which continues to be a public health challenge in tropical countries and Brazil has the second largest number of cases worldwide [2]. Leprosy has different clinical manifestations depending on the host immune response to the bacilli. According to the Ridley and Jopling classification, leprosy is classified into five clinical forms: two polar forms (TT and LL) and three intermediate forms known as borderline forms, that present some features among the polar forms, being classified into borderline-tuberculoid (BT), borderline-borderline (BB) and borderline-lepromatous (BL) [3,4]. In tuberculoid forms (T-Lep), patients present local skin lesions while limited bacillary dissemination is associated with a high cell-mediated immune response. In contrast, in lepromatous forms (L-Lep), patients present disseminated skin lesions with ineffective cell-mediated immunity and intense humoral response, being associated with failure to control bacillary growth and spread among tissues [3–5].

Evidence exists to suggest that genetic factors play a role in the type of host immune response against *M. leprae* [6]. Most people do not develop clinical signs of leprosy, even after sustained exposure to *M. leprae* [7]. Several polymorphisms were associated with leprosy susceptibility, for example, PARK2 and PACRG [8], TLR1 [9–10], TLR2 [11], IFN-γ [12], TNF [13], LTA [14] and IL-10 [15] among other genes [16,17]. Although host genetics is recognized as an important factor for leprosy susceptibility, it is also clear that other biological or environmental factors, such as gender, nutrition, poverty, BCG status and the aging process also contribute to leprosy outcome.

Population aging is a global process observed in both developed and developing countries and is closely related to an increased life expectancy and reduced fertility observed in contemporary history [18]. According to standards used in other developing countries, elderly population in Brazil has been growing since the 90s. The country witnessed 18% growth from 2012 to 2017 and has a registered number of elderly (60 years of age or older) of over 30.2 million people. In 2019, this group represented 13% of the entire population, and, over the upcoming decades, it is expected to grow two-fold. The country is ranked 6[th] in terms of size of the elderly population worldwide [19].

Two recent Brazilian studies showed that leprosy prevalence has decreased in all age ranges, except among the elderly. In addition, there is a significant increase in the frequency of lepromatous male patients age 59 and older [20,21]. Aging is associated with higher incidence of infectious diseases due to a decline in the adaptive immune response, called immunosenescence [22]. Several studies reported age-related changes in the immune system that increase susceptibility to infectious diseases, for example, accumulation of late-stage memory CD8[+] T cells, inversion in the CD4:CD8 ratio, thymus involution and increase in oxidative stress [22–24].

According to the free radical hypothesis and mitochondrial theory, aging can be considered a consequence of the accumulation of biomolecule oxidation damage caused by free radicals and reactive oxygen species (ROS), and the mitochondria is recognized as source and target of this process [25,26]. Thus, these changes generate the accumulation of harmful molecules, such as carbonylated proteins, malondialdehyde and 4-Hydroxy-2-nonenal (4-HNE) [27,28]. Moreover, oxidative damage is linked with immune system disfunction in the elderly since ROS can induce thymic involution, cellular senescence and impair the activation and proliferation of T lymphocytes [22,29,30].

Thus, we hypothesized that elderly leprosy patients have a distinct biochemical and immunological profile associated with immunosenescence that would require personalized care, treatment and follow-up. Oxidative damage might be higher among elderly patients compared to both young and healthy elderly (non-leprosy) individuals. In this study, we investigate the association between oxidative damage and the antioxidant enzyme system in blood and skin samples using a cohort of 112 subjects between 20–89 years of age, separated into six subgroups according to age range and clinical form of leprosy, as well as healthy (non-leprosy) volunteers. Significant increase in oxidative damage in both blood and skin lesions was observed in elderly patients. More interestingly, these results are associated with lower levels of antioxidant enzyme gene expression among elderly L-Lep patients.

## Methods

### Ethics statement

The study was approved by the Institutional Ethics Committee of the Oswaldo Cruz Foundation/FIOCRUZ (permit protocol number 27052919.0.0000.5248). All leprosy patients and healthy participants signed a written consent form to participate in the study. Biological samples from leprosy patients were obtained at the Leprosy clinic (FIOCRUZ/Rio de Janeiro). Healthy (non-leprosy) elderly individuals were recruited by the Human Aging Laboratory, GeronLab, Policlínica Piquet Carneiro (UERJ/Rio de Janeiro).

### Participants and study design

The leprosy patients involved in the study were classified according to Ridley and Jopling criteria (1966) following confirmation of the diagnosis by clinical examination and histopathological analysis of skin lesions. Blood and skin lesion samples were collected before treatment.

Patients were classified according clinical forms and recruited according to age into two groups: young (Y; ranging from 20 to 40 years of age) and elderly (E; over 60 years of age). All patients and healthy volunteers lived in the metropolitan region of the state of Rio de Janeiro —Brazil, a leprosy endemic area. Exclusion criteria for leprosy patients and healthy elderly volunteers included relapse, pregnancy or breast-feeding women, and co-infections such as tuberculosis, hepatitis B and C, and HIV infection. Hypertensive and diabetic individuals under drug control were included.

## Nitric oxide quantification

The quantification of nitric oxide was carried out indirectly through nitrate and nitrite measurement, according to the commercial Nitrate/Nitrite Colorimetric Assay (Cayman Chemicals, Ann Arbor, MI-USA, #780001) in serum samples from healthy controls and patients. Each serum sample was analyzed in triplicate.

## Protein carbonyl group quantification

The carbonyl content of whole serum proteins was measured using the Levine method [31]. A solution of 10 mM 2.4-dinitro-phenylhydrazine (DNPH) was added to serum aliquots and left to react for 1 h in the dark. Thereafter, serum proteins were precipitated with 10% trichloroacetic acid, followed by three washing steps with ethanol/ethyl acetate (1:1 v/v). The pellet was solubilized in 6 M guanidinium hydrochloride at 37°C for 15 min, forming a light-yellow solution. The carbonyl content was determined from the absorbance at 366 nm (molar absorption coefficient, 22.000 $M^{-1}$/cm) using a SpectraMax 190 Spectrophotometer (Molecular Devices, San Jose, CA-USA).

## Immunofluorescence assay

Frozen skin lesion sections assays were performed using a Leica LM3000 cryostat and fixed in paraformaldehyde. Unspecific binding sites were blocked with 10% Fetal Calf Serum (FCS, GIBCO, Life Technologies) in 0.01 M PBS for 1 h at room temperature. Permeabilization was performed by incubation with 0.05% Triton X-100 for 15 min. Rat IgG2b anti-human CD8 (1:50; Abcam, ab60076), mouse IgG2a anti-human CD45RO (1:25; Abcam, ab86080), and rabbit IgG anti-human 4-Hydroxynonenal (1:50, Abcam, ab46545) or their respective isotypes were diluted in 1% Bovine Serum Albumin (BSA, Sigma-Aldrich) in 0.01 M PBS and incubated at 4°C overnight. Tissue sections were washed 3 times and incubated with Alexa Fluor 594 goat anti-Rat IgG (1:1000, Abcam, ab150164), Alexa Fluor 633 goat anti-mouse IgG1 (1:1000, Thermo Fisher Scientific, A-21126) and Alexa Fluor 488 goat anti-rabbit IgG (1:1000, Abcam, ab150077) secondary antibodies for 1:30 h at room temperature. Nuclei were stained with 4′-6-diamidino-2-phenylindole (DAPI; 1:10000, Molecular Probes, D1306), and slides were mounted with VECTASHIELD Mounting Medium (Vector Laboratories, H-1000). Tissues were imaged with an Axio Observer Z1 (Carl Zeiss, Oberkochen, Germany) using an EC Plan-Neofluar 20x/0.50 objective and Plan-Apochromat 63x/1.3 oil objective. Images were acquired with an AxioCam HRm digital camera as confocal images by structured illumination using Apotome (Carl Zeiss) and mathematically deconvoluted by AxioVision Rel. 4.6 software (Carl Zeiss). DAPI fluorescence images were maintained as conventional fluorescence for clarity reasons. For quantitative analysis of CD8+, CD45RO+ and HNE-associated cells, 10 microscopic fields were imaged, and the number of positive cells was counted in each field. The results were obtained from the mean of field counts determined by three independent observers.

## Total RNA extraction and cDNA synthesis

Total RNA from whole blood obtained by venous puncture was isolated using the PAXgene™ Blood RNA kit (Qiagen, Hilden, Germany) in accordance with the manufacturer´s instructions. In the case of biopsy specimens, skin lesion samples (6 mm³ punch) were mechanically lysed using a Polytron Model PT3100 Homogenizer (Kinematica AG, Lucerne, Switzerland) in 2 mL of TRIzol™ Reagent (Thermo Fisher Scientific, Massachusetts, USA) followed by RNA extraction according to manufacturer's instructions. After isolation, total RNA was treated with TURBO™ DNase (Thermo Fisher Scientific). Subsequently, RNA concentration and quality were evaluated using a NanoDrop ND1000 Spectrophotometer (NanoDrop, Wilmington, USA) and integrity and purity were evaluated by 1.2% agarose gel electrophoresis and observation with a UV Transilluminator (Bio-Rad Inc., Hercules, CA, USA). One microgram of total RNA obtained from whole blood and skin biopsy specimens were reverse transcribed into complementary DNA (cDNA) using SuperScript™ VILO™ Master Mix, according to manufacturer's instructions (Thermo Fisher Scientific).

## Gene expression analysis by real time RT-qPCR

Quantitative RT-PCR was carried out in a final volume of 10 μL containing 200 nM of each SYBR green designed primers (S1 Table), 1X Fast SYBR™ Green Master Mix (Thermo Fisher Scientific) and 10 ng of cDNA. All reactions were carried out in triplicate and appropriate negative controls (no reverse transcriptase and no template controls) were incorporated into each run. Briefly, reactions were performed on a StepOnePlus™ Real-Time PCR System (Thermo Fisher Scientific). An initial incubation at 95˚C for 20 seconds was followed by 40 cycles of denaturation at 95˚C for 3 seconds and annealing and extension at 60˚C for 30 seconds. A melt curve stage was performed for each specific amplification analysis (95˚C for 15 seconds, 60˚C for 1 minute, and 95˚C for 15 seconds). The relative expression of the genes of interest was normalized by ribosomal protein L13. Quantitative PCR data analysis was performed by the $N_0$ method implemented in LinRegPCR v. 2020.0, which considers qPCR mean efficiencies estimated by the window-of-linearity method [32,33]. Briefly, $N_0$ values were calculated in LinRegPCR using default parameters. Then, $N_0$ values from each gene of interest (GOI) were normalized by the $N_0$ of the reference gene (REF) *RPL13a* ($N_{0GOI}/N_{0REF}$).

## Biomark Fluidigm gene expression analysis

Gene expression from whole blood was measured using Biomark's microfluidic-based qPCR technology. Briefly, cDNA was obtained from RNA as described above and then 1.25 μL of cDNA (from stock concentration of 5 ng/μL) was pre-amplified with a pool of 96 primer pairs (final concentration of 50 nM) with 1X TaqMan PreAmp Master Mix (Applied Biosystems, USA, # 4391128) in a GeneAmp PCR System 9700 thermocycler for 14 cycles. Pre-amplified cDNA was then diluted 1:5 in TE (10 mM Tris, 0.1 mM EDTA) and stored at—20˚C until the following day. The Biomark Fluidigm reaction was performed in 96.96 Gene Expression Dynamic Array chips (Fluidigm BMK-M-96.96GT) using 95 samples (plus one non-template control) and 96 primers pairs, according to manufacturer's instructions. For the Biomark Fluidigm reaction, 1.7 μL of pre-amplified cDNA was combined with 1X TaqMan Gene Expression Master Mix (Applied Biosystems, USA, #4369016) plus 1.7 μL of each target primer (20 μM), in a final volume of 5.0 μL. Priming, mixing, and cycling procedures were all carried out according to Biomark Fluidigm's protocol. Cycling conditions included an initial incubation at 50˚C for 2 min, followed by 70˚C for 30 min, followed by a UNG and hot start step: 50˚C for 2 min, 95˚C for 10 min. Subsequently, reactions were submitted to 35 PCR cycles of

95˚C for 15 seconds and 60˚C for 1 minute. Finally, a melting curve step was included with temperatures ranging from 60 to 95˚C. For data analysis, initial quality control was performed based on melting curve analysis (MCA) using Fluidigm Real-Time PCR Analysis Software v. 4.5.2, where targets with multiple dissociation curve peaks were removed from further analysis. Raw data were then exported and processed with custom R scripts [34]. In brief, foreground data (Eva Green) was adjusted by subtraction of background (Rox) intensity to generate Rn (background-adjusted accumulated fluorescence). Quantitative PCR reaction efficiency was estimated by fitting a four-parameter sigmoid model according to Rutledge & Stewart, using functions from the R package qpcR v. 1.41–1 [35]. Cycle thresholds (Ct) were determined from the maximum of the second derivative of the fitted sigmoid curve. Cts and efficiencies were used to estimate relative expression based on the method proposed by Pfaffl [36]. The normalization factor used in the denominator for relative expression consisted of the geometric mean from *RPS16*, *RPL13* and *RPL35* genes, selected as the most stable by the R version of the geNorm algorithm [37,38].

## Statistical analysis

Results were analyzed by Statistical Package for the Social Sciences (SPSS) V. 10.1 (SPSS, Inc., Chicago, IL, USA) and GraphPad Prism V. 8 (San Diego, CA, USA) software. After testing for normality (Shapiro-Wilk normality test), non-normally distributed data were analyzed by non-parametric tests. The Mann/Whitney U-test was used to test the differences between two groups, and comparisons between more than two groups were examined by the Kruskal-Wallis test followed by post-hoc Dunn's correction. Friedman test was used to analyze the effects of bacilli elimination on oxidative stress in L-Lep patients. Normally distributed data were compared using one-way ANOVA followed by Tukey's multiple comparisons test. General Linear Model (GLM) was used to test the independent effects of age and bacilloscopic index (BI) on protein carbonyl levels in L-Lep groups. In this procedure, the statistical significance of each factor (age/BI) is controlled for the effect of the second factor.

## Results

### Characteristics of the studied subjects

The present study included 87 leprosy patients and 25 healthy volunteers. Among the patients, 46 (52.9%) were men and 41 (47.1%) were women. Multibacillary leprosy patients (L-Lep) presented two clinical forms: 47.6% (20/42) were borderline lepromatous (BL) and 52.4% (22/42) were polar lepromatous (LL). Among the paucibacillary patients (T-Lep), 13.3% (6/45) were polar tuberculoid (TT) and 86.7% (39/45) were borderline tuberculoid (BT). Patients were stratified according to age (young (Y) or elderly (E)) and clinical forms of leprosy yielding four subgroups while healthy individuals were only stratified into two subgroups by age. Other demographic and clinical characteristics of all subjects studied in this work are shown in Table 1.

The frequency of leprosy reactions was similar between groups; thus, it seems that reactions were not influenced by age. Although the educational level was quite unequal between young (Y) and elderly (E) leprosy patients ($P = 0.0051$ between E T-Lep vs. Y T-Lep; $P = 0.0173$ E L-Lep vs. Y L-Lep), the time for diagnosis of the disease was similar for all subgroups, averaging 12 months from symptom onset to diagnosis. The presence of the BCG scar was more frequent in younger patients and additionally the result from Mitsuda reaction was higher in young T-Lep patients.

**Table 1. Demographic and clinical data of the individuals of this study.**

| | TT/BT >60 ys (E T-Lep) | TT/BT 20–40 ys (Y T-Lep) | BL/LL >60 ys (E L-Lep) | BL/LL 20–40 ys (Y L-Lep) | Healthy volunteers >60 ys (E HV) | Healthy volunteers 20–40 ys (Y HV) |
|---|---|---|---|---|---|---|
| N | 25 | 20 | 20 | 22 | 15 | 10 |
| Age (Mean ± SD) | 69.2 ± 7.2 | 31.8 ± 5.4 | 68.1 ± 6.4 | 31.3 ± 5.9 | 77 ± 6.6 | 29.6 ± 6.2 |
| Gender | | | | | | |
| Male N (%) | 6 (24%) | 12 (60%) | 13 (65%) | 15 (68%) | 7 (47%) | 4 (40%) |
| Female N (%) | 19 (76%) | 8 (40%) | 7 (35%) | 7 (32%) | 8 (53%) | 6 (60%) |
| BI (Mean ± SD) | _ | _ | 3.8 ± 1.3 | 4.3 ± 1.2 | - | - |
| WHO disability grade I or II (%) | 17.40% | 6.20% | 60% | 42.10% | - | - |
| Lepromin Test (Mitsuda mm ± SD) | 6.88 ± 3.99 | 10.07 ± 2.84 | 0 | 0 | - | - |
| History of Reaction (%)* | | | | | | |
| Type 1 N (%) | 4 (18.2%) | 3 (21.4%) | 4 (22%) | 4 (19%) | - | - |
| Type 2 N (%) | 0 | 0 | 5 (27,8%) | 6 (28,5%) | - | - |
| Time to diagnosis** (Mean±SD) | 12.3 ± 9.4 | 11.3 ± 6.5 | 10.7 ± 11.5 | 12.1 ± 7.5 | - | - |
| Presence of BCG vaccine scar N (%) | 7 (28%) | 17 (85%) | 1 (5%) | 11 (55%) | - | - |
| Scholling degree (years) | | | | | | |
| 0–5 | 17 (81%) | 3 (20%) | 13 (86.7%) | 9 (42.8%) | - | - |
| > 5 | 4 (19%) | 12 (80%) | 2 (13.3%) | 12 (57.2%) | - | - |

* Up to two years after diagnosis. SD: standard deviation.

** Time in months between onset of symptoms and clinical diagnosis.

## Increased levels of protein carbonyls in serum samples of elderly leprosy patients

To check oxidative damage, serum samples from patients and healthy individuals were analyzed as to the production of nitric oxide (NO) and to the levels of protein carbonyls. The concentration of NO in the serum did not show significant differences between young and elderly leprosy patients, regardless of the clinical form of the disease. The NO concentration was significantly higher in young volunteers than in elderly ones ($P = 0.0245$ Fig 1A).

Protein carbonylation is a major form of protein oxidation and is used as a parameter of oxidative stress. Serum samples of all the elderly leprosy patients demonstrated a significant increase in protein carbonyl levels. Furthermore, when elderly and young individuals were compared, the difference was only significant for L-Lep patients and for the elderly HV group (Fig 1B–1D). Among T-Lep patients, the protein carbonyl concentration in serum samples was also higher in elderly individuals than in the young, but no significant difference was observed between groups ($P = 0.1223$; Fig 1C). Serum samples of all the elderly leprosy patients demonstrated a significant increase in protein carbonyl levels ($P = 0.0044$; Y Lep vs. E Lep; Fig 1E). Although healthy elderly volunteers have a significant increase in the concentration of protein carbonyls compared to healthy young ones, in our study both elderly L-Lep and T-Lep leprosy patients have a higher concentration of protein carbonyls than healthy elderly volunteers ($P = 0.0257$ and $P = 0.0291$, respectively; Fig 1E).

In elderly T-Lep patients, the higher levels of carbonylated proteins may be associated with higher *NOX1* gene expression, which is responsible for the production of superoxide anion. Moreover, the normalized expression values of *NOX1* enzyme in blood samples was higher in E T-Lep patients compared to Y T-Lep patients ($P = 0.0005$; S1A Fig). As expected, the number

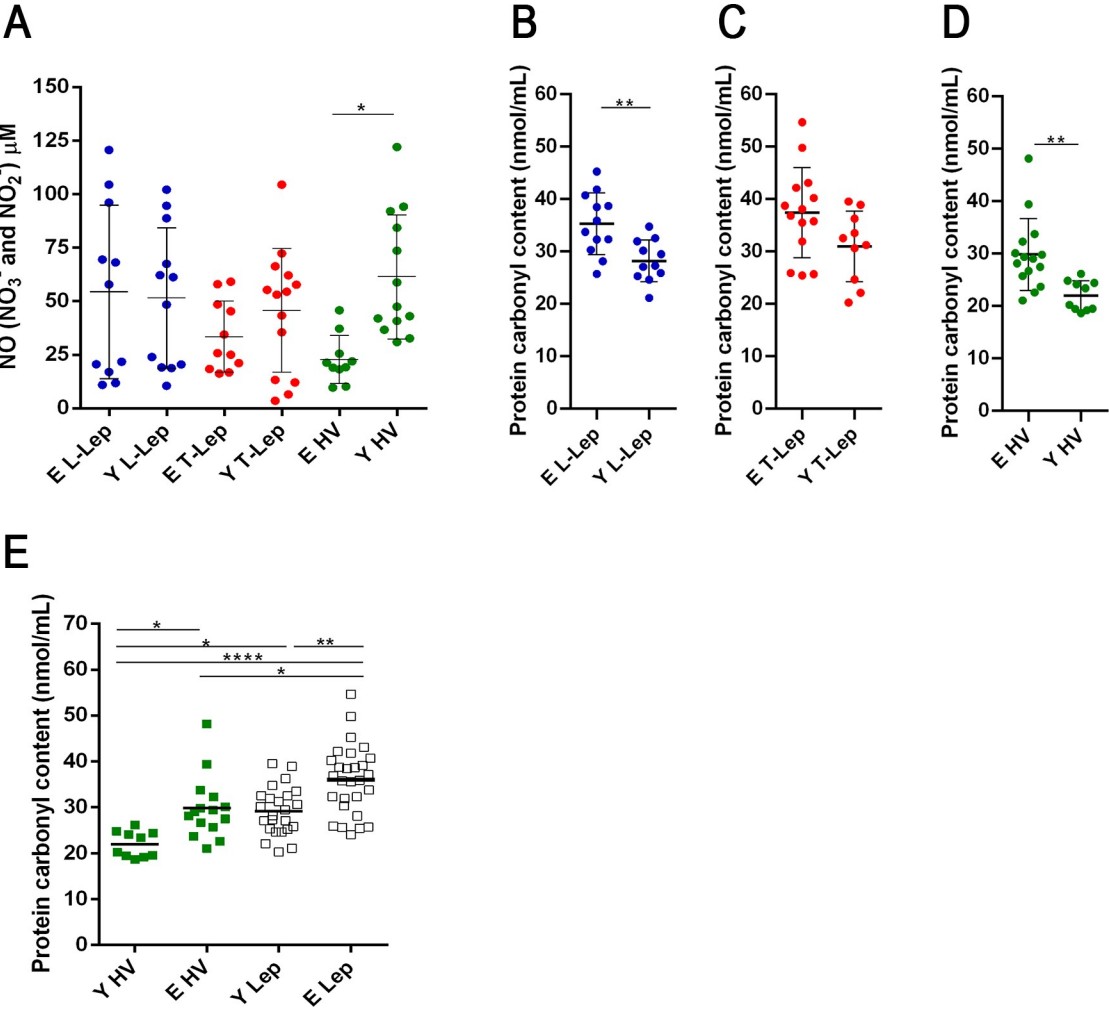

**Fig 1. Evaluation of oxidative stress in serum samples. (A)** Concentration of NO in the serum of patients and healthy volunteers. Horizontal short lines represent mean of each group. Analysis of variance was performed by one-way ANOVA followed by Tukey's multiple comparison test. Graphs represent means ± SD of serum protein carbonyl group concentration and a Mann/Whitney U-test for continuous variables was performed to evaluate significant differences between groups of **(B)** L-Lep patients (Y n = 12 and E n = 13), **(C)** T-Lep patients (Y n = 12 and E n = 15) and **(D)** healthy volunteers (Y n = 10 and E n = 15), each circle represents an individual. **(E)** Serum protein carbonyl group concentration, each square represents an individual (same n above) and the short horizontal lines represent means. Analysis was performed by one-way ANOVA followed by Tukey's multiple comparison test. $^*P < 0.05$, $^{**}P < 0.01$ and $^{****}P = 0.0001$. Abbreviations: E—Elderly; Y—Young; T-Lep (paucibacillary; TT/BT patients); L-Lep (multibacillary; BL/LL patients); HV—healthy volunteers.

of men and women was uneven between elderly L-Lep and T-Lep patient groups. In the GLM analysis of protein carbonyl levels among L-Lep patients, BI was not statistically significant (beta = - 0.762; 95% IC = [-2.405; 0.881], (P = 0.346), while age was an independent factor associated with protein carbonyl level (beta = 6.867; 95% IC = [2.500; 11.234], (P = 0.004). Therefore, the increased concentration of protein carbonyl is related to the aging process; however, damage is not influenced by bacillary load in L-Lep patients.

## Protein carbonyl concentration increases one year after multidrug therapy (MDT) discharge in young L-Lep patients

After evaluating the concentration of protein carbonyls in newly diagnosed leprosy patients before beginning specific disease treatment (MDT), we conducted a follow-up study to

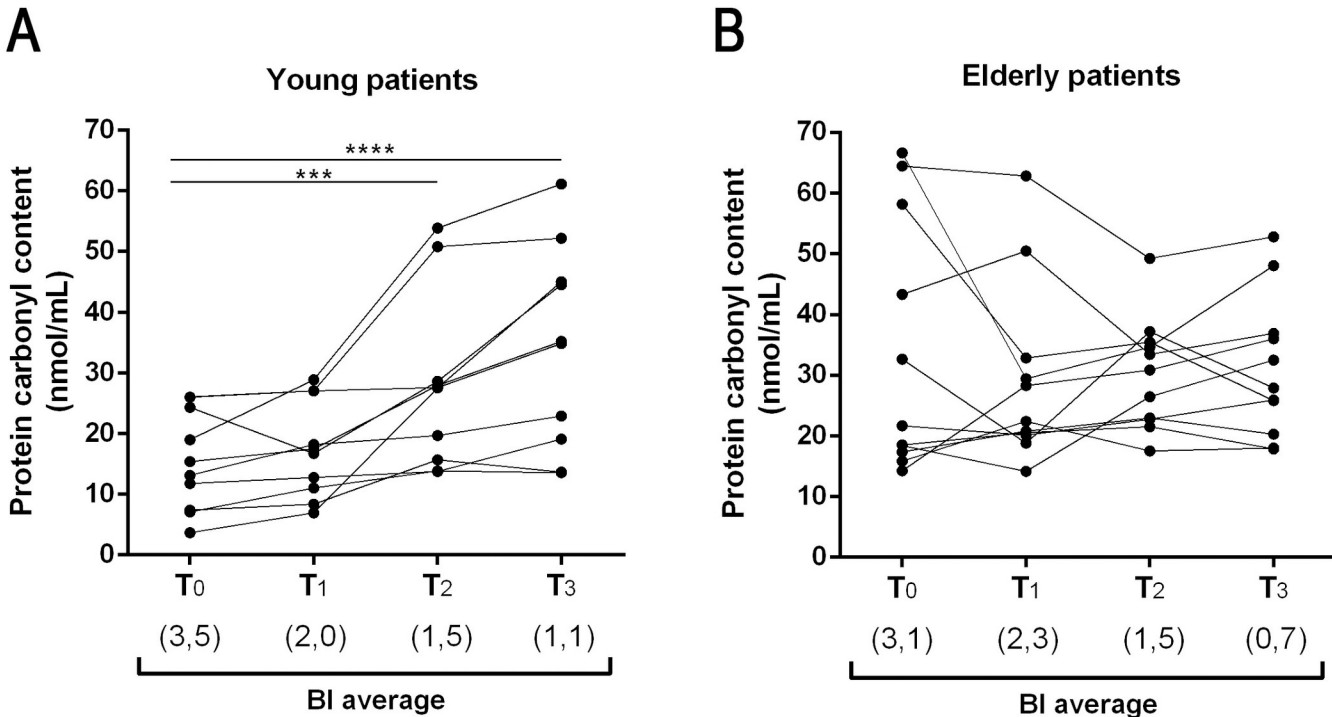

**Fig 2. Increase in serum protein carbonyl concentration one year after multidrug therapy (MDT) discharge in the young L-Lep group. (A)** Effects of treatment on oxidative stress in Y L-Lep patients (n = 9) and **(B)** in E L-Lep patients (n = 10): T0 –at diagnosis, before starting MDT; T1—discharge after treatment; T2—one year after discharge; T3—two years after discharge. Each circle represents a patient and lines represent the follow up. The Friedman test was performed to evaluate significant differences between the beginning of MDT and the years following discharge. Values in parenthesis refer to the BI average for each time point. ***$P < 0.001$ and ****$P = 0.0001$.

understand the relationship between variation in bacillary load and oxidative stress after completion of treatment. Thus, patient sera were obtained at four different time points: untreated patients before the start of MDT (T0), at MDT discharge (T1), one year after MDT discharge (T2), and two years after MDT discharge (T3; Fig 2A and 2B). Patients affected by acute inflammation (leprosy reactions) at the time points were removed from this analysis. In both young and elderly L-Lep patients, we observed a reduction in the mean bacterial index (BI) after MDT (from 3.5 to 1.1 in young, and from 3.1 to 0.7 in elderly leprosy patients). Although BI decreased in both groups, the concentration of protein carbonyls increased in young patients only, while a more heterogeneous profile was observed in elderly leprosy patients, where higher levels where detected before and after MDT.

### Elderly leprosy patients present higher presence of 4-HNE in skin lesion samples

To investigate oxidative damage caused by ROS in skin lesions of leprosy patients, cutaneous fragments of these tissues were stained with anti-4-HNE (a marker of oxidative tissue damage), anti-CD8, and anti-CD45RO antibodies. We traced anti-CD8 and anti-CD45RO antibodies because of the accumulation of memory CD8+ T cells (CD8+CD45RO+) in tissues as a result of aging. Thus, to assess oxidative stress in the surroundings of these cells involved in skin immunosenescence, we immunolocalized this cell population and 4-HNE in skin lesions of four patients from each group (Fig 3A). Regardless of the clinical form of the disease, the percentage of 4-HNE-associated cells was significantly higher in the elderly inflammatory infiltrate than

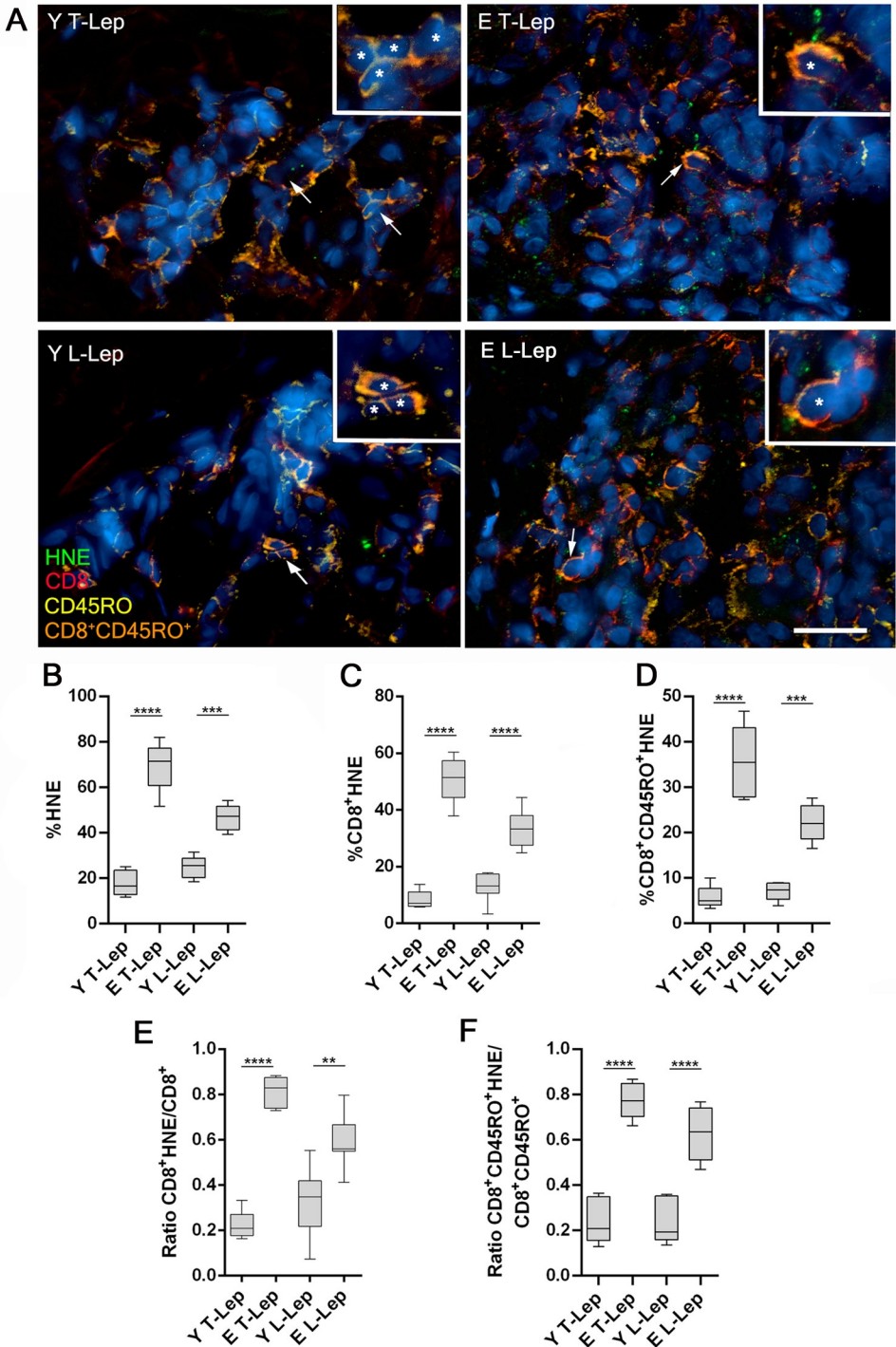

**Fig 3. Analysis of oxidative stress by quantifying 4-hydroxynonenal in skin lesions of leprosy patients.**
Immunofluorescence assays were performed to determine the number of memory CD8+ T cells (CD8+CD45RO+)
associated with 4-hydroxynonenal (HNE) signal, as well as HNE abundance in tissue. (**A**) Representative images of
skin lesions from a young paucibacillary patient (Y T-Lep), an elderly paucibacillary patient (E T-Lep), a young
multibacillary patient (Y L-Lep) and an elderly multibacillary patient (E L-Lep), presenting HNE (green, Alexa Fluor
488), CD8 (red, Alexa Fluor 594), CD45RO (yellow, Alexa Fluor 633) and nuclei stained by DAPI (blue). Arrows and
asterisks indicate CD8+CD45RO+ cells, stained in orange. In the enlarged inserts, more intense HNE signal in elderly
T-Lep and L-Lep patients is observed. Scale bar represents 40µm in main image and 20 µm in inserts. The graphs
present frequency of (**B**) HNE-associated cells, (**C**) double-positive CD8+HNE-associated cells, (**D**) triple-positive

CD8$^+$CD45RO$^+$HNE-associated cells (**E**) ratio between frequency of CD8$^+$HNE-associated cells and total CD8$^+$ cells and (**F**) ratio between frequency of CD8$^+$CD45RO$^+$HNE-associated cells and CD8$^+$CD45RO$^+$. Results are reported as percentage of positive cells in interquartiles (minimum to maximum) in groups of 4 patients. These data were compared using one-way ANOVA followed by Tukey's multiple comparisions test. $^{**}P < 0.005$, $^{***}P < 0.001$ and $^{****}P < 0.0001$.

in young patients (Fig 3B). Similarly, the CD8$^+$ HNE-associated T cells were significantly increased among elderly, when compared to young individuals (Fig 3C). Furthermore, the frequency of HNE-associated memory CD8$^+$ T lymphocytes (CD8$^+$CD45RO$^+$HNE-associated) was significantly higher in elderly patients when compared to young patients (Fig 3D). Moreover, the ratio between CD8$^+$HNE-associated cells and CD8$^+$ cells was higher in E L-Lep than in Y L-Lep patients ($P = 0.012$; Fig 3E). This ratio was also significantly higher in E T-Lep when compared to Y T-Lep patients ($P < 0.0001$; Fig 3E). Furthermore, the ratio between memory CD8$^+$ T lymphocytes associated to 4-HNE signal and memory CD8+ T cells was significantly higher in elderly patients than in young ones, regardless of the clinical form of leprosy ($P < 0.05$; Fig 3F). In the same figure, it is possible to note that around 80% of memory T lymphocytes from skin lesions in elderly T-Lep patients showed oxidative damage as measured by the presence of 4-HNE. So, it was also possible to note an increased oxidative damage in skin lesions from elderly patients, in addition to the damages observed in the blood samples.

## Lower levels of antioxidant enzymes in elderly L-Lep skin lesion

To investigate gene expression of antioxidant enzymes, real time RT-qPCR assays were performed in skin lesion specimens using a StepOnePlus™ Real-Time PCR System. Gene expression of superoxide dismutase 1 (*SOD1*) and 2 (*SOD2*), glutathione-disulfide reductase (*GSR*), and glutathione peroxidase 1 (*GPX1*) enzymes can contribute to better understanding of the imbalance between ROS production and the role of the antioxidant defense system in the individuals involved in this study. Regarding enzyme gene expression in skin lesion samples, the normalized expression values were remarkably similar between young and elderly T-Lep patients. Gene expression levels of antioxidant enzymes in the skin were significantly higher in young L-Lep patients when compared to the elderly, such as SOD1 and SOD2 enzymes (Fig 4A and 4B; $P = 0,0367$ and $P = 0,0485$ respectively). Only *GSR* did not show a significant increase in young L-Lep patients ($P = 0.4144$), although the normalized expression values were also higher in these patients (Fig 4C). In these experiments, the *GPX1* gene stood out from the rest due to significant upregulation in young L-Lep patients in relation to elderly ($P < 0.0001$; Fig 4D). Measured by Biomark's Fludigm technology, the expression of antioxidant enzyme genes in blood samples was quite similar in both groups of patients across all clinical forms of leprosy and healthy volunteers respectively (Fig 4E–4H). Increased expression of these antioxidant enzymes in Y L-Lep patients was associated with increased ROS production. Furthermore, normalized expression values of *NOX2* in skin lesions were higher in Y L-Lep patients when compared to E L-Lep patients ($P = 0.0039$; S1B Fig). So, according to the results, young L-Lep patients presented an increased gene expression of antioxidant enzymes in the skin samples, when compared to elderly L-Lep patients.

## Discussion

Infections are favored by age-related alterations in immune response with a consequent increase in susceptibility and disruption in cytokine production [39–41]. In leprosy, age-related mechanisms of alterations affecting the immune system still require clarification since most individuals present the signs and symptoms of the disease between the second and the

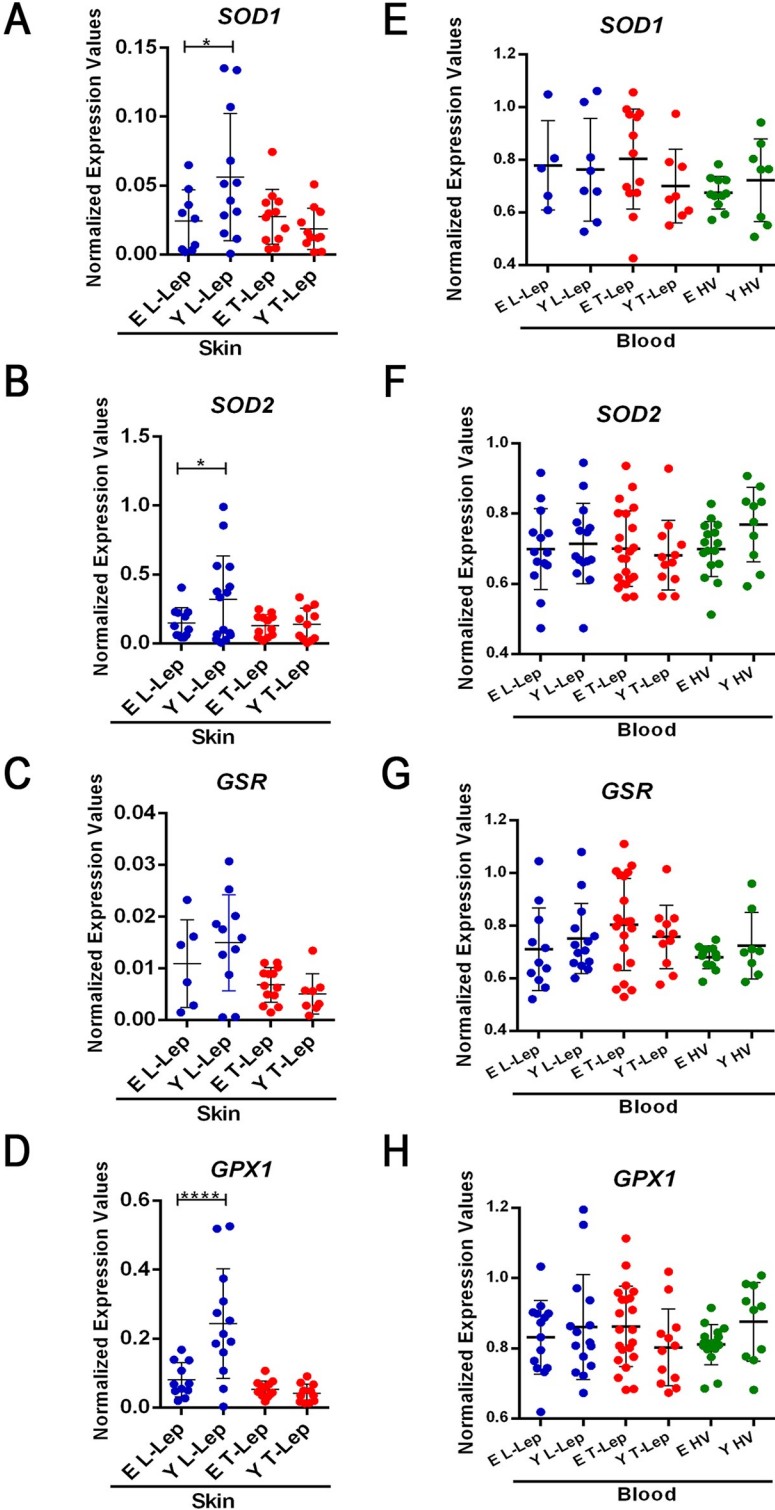

**Fig 4. Gene expression analysis of antioxidant enzymes in whole blood and skin lesions.** Quantitative PCR (qPCR) evaluation of mRNA levels of SOD1 (**A** and **E**), SOD2 (**B** and **F**), GSR (**C** and **G**) and GPX1 (**D** and **H**) was performed in whole blood and skin lesion samples. Each circle represents one individual. Horizontal bars represent the mean values ± SD. Data analysis was performed using ordinary one-way ANOVA followed by Holm-Sidak's multiple comparison test (**A**, **B**, and **G**). Kruskal-Wallis test followed by Dunn's multiple comparison post-test was used for the analysis of non-normally distributed and/or heteroscedastic data (**C**). Variation in number of individuals for each gene

expression analysis was due to lack of amplification. $^{**}P < 0.01$, $^{***}P < 0.001$ and $^{***}P = 0.0001$. Abbreviations: E–Elderly; Y–Young; T-Lep–TT/BT patients; L-Lep–BL/LL patients.

fourth decades of life [5]. In view of the above, we hypothesize whether oxidative stress patterns could be detected among elderly patients, which could indicate specific immunophenotypic profiles potentially affecting severity to leprosy or even the course of the disease. Here, we provide evidence that oxidative stress, as detected by carbonylated proteins in sera, is higher in elder patients as compared to younger patients or elderly healthy individuals. To corroborate these findings, we also detected higher levels of tissue damage induced by oxidation in skin samples, although no differences were observed between L-Lep and T-Lep patients.

Part of the decline in physiological function during the natural aging process is linked to the imbalance between the production of ROS and the antioxidant defense system [42,43]. Damage caused by ROS can be measured by quantifying certain molecules, such as 4-HNE and carbonyl proteins. Our data showed that oxidative stress is higher in elderly leprosy patients compared to healthy elderly subjects. Furthermore, marked oxidative damage in L-Lep patients does not seem to be linked to bacillary load, at least when comparing elderly and young patients. The relationship between *M. leprae* and the increased damage caused by ROS remains unclear, but previous studies have suggested a correlation between bacillary load and oxidative stress [44,45]. However, these works did not clarify the mechanism of such process during aging.

Our data demonstrate that leprosy may reduce oxidative stress among young leprosy patients, considering that after treatment and reduction in bacillary load, the concentration of protein carbonyls increased in the sera of these patients. In view of the above, some studies have already shown that there is an increase in ROS production during the first months of multidrug therapy (MDT), probably due to dapsone [46,47]. In the case of young L-Lep patients, oxidative stress increased one year after release from MDT while, in elderly patients, oxidative stress remained high both during and years after leprosy treatment. From demonstrated that, in young L-Lep patients, progressive *M. leprae* elimination may explain the increase in carbonylated protein levels over the years following treatment, considering the increased mRNA expression of antioxidant defense enzymes (*SOD1*, *SOD2*, and *GPX1*) in these unattended patients. Similar findings have been reported by Medeiros and collaborators who showed that *M. leprae* is capable of reducing oxidative stress by increasing the enzymatic activity of glutathione reductase and reducing mitochondrial activity in the Schwann cell model. Thus, the leprosy bacillus seems to be able to modulate oxidative stress, whereas this reduction is strategic for promoting an appropriate niche for their survival [48]. Genetic studies have identified several genes and genomic regions contributing to the control of host susceptibility to leprosy. Our finding of low levels of SOD2 gene expression in skin samples of elderly L-Lep patients is interesting, considering that variants of this gene are located at the same genomic locus as *PRKN*, a well-known leprosy susceptibility gene [49]. Future research may better explain the exact mechanism of interaction between the pathogen and expression of antioxidant enzyme genes in other tissues, such as blood and skin.

Our data suggest that high levels of 4-HNE in skin lesions of elderly L-Lep patients are linked with low antioxidant enzyme expression. The enzyme Glutathione peroxidase 1 (GPx1) catalyzes the conversion of $H_2O_2$ into water using glutathione as a substrate and also promotes the reduction of lipid peroxides, such as 4-HNE [50]. Therefore, higher GPX1 gene expression in skin lesions can protect against oxidative damage in young L-Lep patients. Additionally, 4-HNE has a harmful effect of amplifying inflammation and increasing tissue damage. These effects are produced from activation of a TLR4/NF-κB-dependent pathway [51].

Previous data showed that *M. leprae* modulates host lipid metabolism to facilitate its survival and reduce the immune response [52,53]. Patients in this study did not present dyslipidemia and there were also no significant differences in the levels of total cholesterol, HDL, LDL or triglycerides between elderly and young patients with same clinical form (S2 Table). A recent study showed that in L-Lep patients, oxidative stress was enhanced due to changes in the chemical composition of HDL, which impairs its antioxidant and anti-inflammatory action [54]. Although an increase in HDL levels in multibacillary patients was not observed in our study, the analysis of changes in the chemical composition of HDL among elderly and young leprosy patients can help to understand the influence of the aging process on the relationship between changes in lipid metabolism of the host and oxidative stress.

The downregulated expression of GPX1 observed in elderly L-Lep patients in this study may be associated with induction of the cell senescence process. ROS, especially $H_2O_2$, which is neutralized by the GPx1 enzyme, alter the regulation of protein expression, such as p53, p21 and p16INK4a, which induce cellular senescence [55,56]. High levels of lipid peroxidation found in skin lesions of elderly leprosy patients, especially in memory CD8$^+$ T lymphocytes, indicate that bacilli elimination can be reduced due to the cellular senescence process that these lymphocytes may be undergoing. In addition, these senescent cells increase tissue damage caused by inflammation due to senescence-associated secretory phenotype (SASP), which is characterized by increased expression and secretion of pro-inflammatory cytokines and chemokines [57,58]. These effects can be amplified through factors of the secretory phenotype and induce the generation of senescence in normal neighboring cells in a paracrine manner.

In addition, higher gene expression of antioxidant enzymes in skin lesions of Y L-Lep patients may be related to oxidative burst. Although these patients present increased gene expression of the Nox2 enzyme, oxidative damage seems to be controlled by the high expression of antioxidant enzymes, mainly glutathione peroxidase. Nox2 is highly expressed in phagocytes and contributes to killing of intracellular pathogens. This enzyme transports electrons from cytoplasmic NADPH to extracellular or phagossomal oxygen to generate superoxide anion [59]. Recent work showed that *Mycobacterium tuberculosis* survived in Nox2-KO macrophages, and high levels of ROS induced via NOX2 were correlated with more favorable tuberculosis treatment outcome [60] Moreover, another recent study showed that a genetic variation in NCF2, a Nox2 complex activator, contributes to the susceptibility to tuberculosis in a Chinese population [61]. These data reinforce the possibility that Nox2 activity is essential to *M. tuberculosis* phagolysosomal degradation. Likewise, Nox2 expression could play a crucial role in eliminating bacillus from the elderly L-Lep patients studied herein.

Under chronic inflammatory conditions, ROS reduce activation signals to the T cell and impair the immune response against pathogens [29]. Despite the regular levels of TCR expression in T lymphocytes isolated from the peripheral blood of patients with chronic diseases, such as leprosy, there are structural changes in the TCR chain, specifically on T-cell receptor ζ-chain (TCRζ). ROS can induce these structural modifications of the TCR, therefore impairing the T lymphocytes activation and proliferation [62]. Thus, the increased ROS production and a deficient antioxidant system are responsible for the induction of hyporesponsiveness in T lymphocytes. In addition, Zea and colleagues demonstrated that alterations on TCRζ expression were correlated with lower levels of IFN-γ release in leprosy patients [63]. In this regard, in L-Lep patients, there is extensive replication of *M. leprae* in macrophages. This reduction in macrophage microbicidal activity has been attributed to the hyporesponsiveness of T cells, as shown in blood and cutaneous lesions of L-Lep patients [64,65]. In addition to the effects of oxidative damage in the suppression of the T cell immune response, there is evidence that the increase in oxidant stress is associated with mycobacterial survival in macrophages. Indeed,

Oberley-Deegan and coworkers showed that MnTE-2-PyP, a ROS scavenger, reduces intracellular *Mycobacterium abscessus* numbers by enhancing phagosome–lysosome fusion [66].

Considering the lower synthesis ability of antioxidant enzymes in elderly L-Lep patients verified in this work, there are various mechanisms used by the cells to repair damages produced by the oxidative stress in leprosy. The co-supplementation of some antioxidants, in addition to the specific treatment for the disease was previously described as a contributing factor to improve such repair and to mitigate oxidative damage, particularly in L-Lep patients. The list of antioxidants tested encompasses zinc [67], vitamin E [68] and ascorbic acid [69]. Most works disclosed a reduction in oxidative damage following supplementation with the antioxidant agent. Nevertheless, none of them addressed elderly patients. Therefore, the administration of an antioxidant as a co-supplement throughout the clinical course of leprosy appears to be a possible alternative to reduce, or even to prevent, oxidative stress. This supplementation could be concomitant to MDT, and even administered after treatment, particularly in elderly L-Lep patients. We have grouped T-Lep (TT/BT) and L-Lep (LL/BL) patients since clinical, immunological and bacteriological features are similar facilitating comparisons of the groups. Also, we decided to fix the recruitment trying to polarize young (from 20-40y/o) and elderly (>60y/o) patients. All these strategies were performed to facilitates interpretation of the data, although we understand that these approaches can bias the results. Studies increasing recruitment (age ranges) and sample size allowing comparisons between clinical forms are necessary to confirm the data.

Briefly, our study showed exacerbated oxidative damage in elderly leprosy patients, when compared to younger patients. This damage noted both in blood and in skin lesions, was higher in the skin of elderly L-Lep patients. This finding possibly arises, at least in part, from the role of ageing in reducing the antioxidant enzyme system in these individuals. Thus, considering that all elderly patients involved in this study have always lived in an endemic area of leprosy, the exacerbated oxidative stress during the aging process can be an important factor for leprosy susceptibility and immunopathogenesis of the disease after the sixth decade of life.

## Supporting information

**S1 Fig. Gene expression of NADPH oxidase enzymes in whole blood and skin lesions.**
Quantitative PCR (qPCR) evaluation of NOX1 mRNA levels (**A**) in whole blood (L-Lep patients, Y n = 12 and E n = 13; T-Lep patients Y n = 12 and E n = 15; HV Y n = 10 and E n = 15), and NOX2 (**B**) in skin lesion samples (Y L-Lep n = 13 and E L-Lep n = 10; Y T-Lep n = 10 and E T-Lep n = 12). Bar graphs represent means ± SD of each group. Data analysis was performed using Kruskal-Wallis test followed by Dunn's multiple comparison post-test. $^{**}P < 0.01$, and $^{***}P < 0.001$. Abbreviations: E–Elderly; Y–Young; T-Lep–TT/BT patients; L-Lep–LL/BL patients; HV–healthy volunteers.
(TIF)

**S1 Table. Oligonucleotides used in the study.**
(DOCX)

**S2 Table. Biochemical data of analyzed group.**
(DOCX)

## Acknowledgments

We are grateful to all the leprosy patients and non-leprosy volunteers for agreeing to participate in the study, as well as to Cristiane Domingues and José Augusto da Silva for the

administrative assistance. Our recognition to Anna Beatriz Robottom Ferreira, a native speaker for editing the text.

## Author Contributions

**Conceptualization:** Danuza Esquenazi.

**Data curation:** Pedro Henrique Lopes da Silva, Danuza Esquenazi.

**Formal analysis:** Pedro Henrique Lopes da Silva, Mayara Abud Mendes, Thyago Leal Calvo, Mariana de Andréa Vilas-Boas Hacker, Flávio Alves Lara, Danuza Esquenazi.

**Funding acquisition:** Euzenir Nunes Sarno, Roberto Alves Lourenço, Milton Ozório Moraes, Danuza Esquenazi.

**Investigation:** Pedro Henrique Lopes da Silva, Milton Ozório Moraes, Flávio Alves Lara, Danuza Esquenazi.

**Methodology:** Pedro Henrique Lopes da Silva, Katherine Kelda Gomes de Castro, Mayara Abud Mendes, Thyago Leal Calvo, Júlia Monteiro Pereira Leal, José Augusto da Costa Nery, Euzenir Nunes Sarno, Roberto Alves Lourenço.

**Project administration:** Danuza Esquenazi.

**Resources:** Milton Ozório Moraes, Flávio Alves Lara, Danuza Esquenazi.

**Supervision:** Danuza Esquenazi.

**Validation:** Milton Ozório Moraes, Flávio Alves Lara, Danuza Esquenazi.

**Visualization:** Pedro Henrique Lopes da Silva, Danuza Esquenazi.

**Writing – original draft:** Pedro Henrique Lopes da Silva, Mayara Abud Mendes, Milton Ozório Moraes, Flávio Alves Lara, Danuza Esquenazi.

**Writing – review & editing:** Milton Ozório Moraes, Danuza Esquenazi.

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
