## [Decision Letter · Decision Letter 0]

8 Dec 2020

Dear Dr. Esquenazi,

Thank you very much for submitting your manuscript "Increased oxidative stress in elderly leprosy patients is related to age but not to bacillary load " for consideration at PLOS Neglected Tropical Diseases. As with all papers reviewed by the journal, your manuscript was reviewed by members of the editorial board and by several independent reviewers. The reviewers appreciated the attention to an important topic. Based on the reviews, we are likely to accept this manuscript for publication, providing that you modify the manuscript according to the review recommendations. 

Sincerely,

Vinicius M Fava, PhD

Associate Editor

Gerd Pluschke

Deputy Editor

Reviewer's Responses to Questions

**Key Review Criteria Required for Acceptance?**

**Methods**

-Are the objectives of the study clearly articulated with a clear testable hypothesis stated?

-Is the study design appropriate to address the stated objectives?

-Is the population clearly described and appropriate for the hypothesis being tested?

-Is the sample size sufficient to ensure adequate power to address the hypothesis being tested?

-Were correct statistical analysis used to support conclusions?

-Are there concerns about ethical or regulatory requirements being met?

Reviewer #1: Methods are mostly adequate; two major concerns regarding the use of the Ridley & Jopling classification system and the definition of the non-affected sample are described in the summary/general comments.

Reviewer #2: -Are the objectives of the study clearly articulated with a clear testable hypothesis stated?

Yes

-Is the study design appropriate to address the stated objectives?

Yes

-Is the population clearly described and appropriate for the hypothesis being tested?

Yes

-Is the sample size sufficient to ensure adequate power to address the hypothesis being tested?

Yes

-Were correct statistical analysis used to support conclusions?

Yes

-Are there concerns about ethical or regulatory requirements being met?

Yes

**Results**

-Does the analysis presented match the analysis plan?

-Are the results clearly and completely presented?

-Are the figures (Tables, Images) of sufficient quality for clarity?

Reviewer #1: Results are interesting, well presented in the text, tables and figures. Minor suggestions are presented in the summary/general comments.

Reviewer #2: -Does the analysis presented match the analysis plan?

Yes

-Are the results clearly and completely presented?

Not always (see Editorial and Data Presentation Modifications section)

-Are the figures (Tables, Images) of sufficient quality for clarity?

Not always (see Editorial and Data Presentation Modifications section)

**Conclusions**

-Are the conclusions supported by the data presented?

-Are the limitations of analysis clearly described?

-Do the authors discuss how these data can be helpful to advance our understanding of the topic under study?

-Is public health relevance addressed?

Reviewer #1: Conclusions are adequate, well supported by the data and well discussed. Minor suggestions to enrich the discussion is presented in the summary/general comments.

Reviewer #2: -Are the conclusions supported by the data presented?

Yes

-Are the limitations of analysis clearly described?

No. Maybe the authors could highlight the relative imbalance between each patient subgroup.

-Do the authors discuss how these data can be helpful to advance our understanding of the topic under study?

-Is public health relevance addressed?

**Editorial and Data Presentation Modifications?**

Reviewer #1: A few suggestions to improve the organization of the manuscript are described in the summary/general comments.

Reviewer #2: Overall, I have minor comments that should be addressed to increase rigor and clarity.

A few typos need to be corrected (e.g. line 125, 126).

The order in table and figure calling is often not correct (e.g. Table S2 appears before Table S1, Fig 1E before Fig 1B…), which disturbs the flow of the article.

- Introduction : 

o please give more details about the borderline forms of leprosy

o Line 123: is it possible to provide a more recent review or article?

o Line 128: a hypothesis should not be conditional (whether…)

- Methods:

o Hypertensive and diabetic individuals were not excluded. Is there any data on ROS production in such individuals?

o Line 257: Graph Prism should be GraphPad Prism if I’m not mistaken

- Results:

o Overall, barplots should be replaced by graphs displaying the distribution of the samples (by box plots or else). Also, these graphs should be presented with the same categories and the same ordering for clarity (e.g. following what’s displayed on Fig 1A). In addition, I think most graphs may display the mean and not the median as it is stated. The use of distinct colors in graphs differentiating L-Lep and T-Lep individuals (and controls) would be of great help for clarity and interpretation.

o Characteristics of the subjects:

The authors compared T-Lep vs L-Lep but have data separating T-Lep (TT and BT) and L-Lep (BL and LL). Why the authors do not take advantage of this classification (which could lead to more accurate conclusions)? 

Table 1: What is the age stratification among TT, BT, BL and LL samples?

o Increased oxidative damage in serum samples of elderly leprosy patients

The paragraph starts abruptly with a conclusion, not stating the hypothesis, and how it was tackled

Line 288: I observe the exact contrary to the statement made by the authors (Fig 1A)

Line 305: Only E T-Lep are different from the other groups: 24-76% vs 65-35%. Could the authors give an explanation to this?

Line 310: Please, in addition to a P value, provide beta (or OR) and CI. 

o Protein carbonyl concentration increases one year after MDT discharge in young L-Lep patients

Why focusing only on L-Lep patients? I would like to see the parallel with T-Lep patients (and healthy controls?). 

Line 326: changes after MDT do not seem “subtle” for each sample. 

o Elderly leprosy patients present higher presence of 4-HNE in skin lesion samples:

What is the conclusion of the paragraph? What to conclude about Figure 3?

o Lower levels of antioxidant enzymes in elderly L-Lep skin lesion:

Fig4 A and B are not commented. 

What about SOD2 results? 

Again, what is the conclusion?

- Discussion:

o Line 404: which antioxidant? GPX1 only? (maybe SOD2?)

o Line 423: The levels of triglyceride in E T-Lep (close to E HV) and Y T-Lep seems quite different to me. Could the author test this? Which comparison and which test was performed in Table S1 ?

o Paragraph starting Line 442: This link between MTB and M. leprae is too implicit and does not bring insights to the point of the paper. Please develop.

o Paragraph starting Line 454: Again, the statements in this paragraph are too implicit. Please develop. 

o Line 463: The authors mention the lower synthesis ability of antioxidant enzymes in E L-Lep but I also see similar levels in E T-Lep. Could the authors comment on this observation?

o The last paragraph is quite confusing (especially line 477 and 478). How to draw a conclusion of L-Lep patients from an observation in T-Lep patients?

**Summary and General Comments**

Reviewer #1: The manuscript describes an investigation of the role of oxidative stress on leprosy pathogenesis in elderly individuals as compared to young affected and a sample of non-affected. Results suggest that the increased oxidative damage related to aging may alter leprosy pathogenesis; the authors suggest that antioxidant therapy may benefit elderly leprosy patients.

The well written manuscript reports interesting and relevant research much needed in leprosy, still a major public health burden in several countries. Several of the authors have an extensive history of contribution to the advance on the understanding of the molecular basis of leprosy pathogenesis, most of whom based at a highly reputed research center with optimal infrastructure for high quality basic research. The results are interesting, add considerably to the body of knowledge in leprosy and are of potential interest of the readers of PLOS NTD.

Upon careful reading of the manuscript, the following few major and minor concerns were raised, to which comments by the authors will be greatly appreciated:

Major concerns:

1. The authors claim to have performed the Ridley & Jopling classification, and the results are used to stratify leprosy patients into groups for comparison; however, this is very difficult today given that several of the parameters originally described by R&J are no longer available, such as the Mitsuda reaction, for example. Can the authors comment on the criteria used to define patients as displaying LL, BL, BT and TT leprosy?

2. There is no description of the sub-sample of non-affected individuals, described by the authors as “healthy volunteers”. A healthy volunteer may be considerably different than a leprosy-free individual, and this may be particularly important if the variables involved (oxidative stress) may be impacted by several types of diseases and conditions. Can the authors kindly expand on how has this sub-sample been characterized?

3. The finding of different levels of SOD2 expression in skin samples of young vs. elder lepromatous patients is interesting considering that variants of this gene, located at the same genomic locus as PARK2 – a well-known leprosy susceptibility gene - have been previously associated with leprosy (Ramos et al., J. Inf. Dis, 2016); please consider using that information in the discussion.

Minor comments:

1. Sometimes the reference to young and elder gets confusing as if refers to leprosy affected patients or non-affected. Please consider revise and if necessary, clarify throughout the text;

2. The authors often include p-values in the text, which is quite uninformative if not associated with the actual values of the parameters under comparison; please consider including to increase the information content;

3. Line 294, the sentence would be clearer if stating that “… (Fig1B-1D) for L-Lep leprosy patients and for the HV group.”

4. Some parts of “results” seem to be better fitted into discussion (for example – and not limited to - lines 305-311 and 327-329), please consider reorganizing;

5. There are a few minor typos along the manuscript, easy to fix upon a careful revision.

Reviewer #2: Leprosy is an infectious disease cause by M. leprae and remains a public health problem in several countries including Brazil. Lopes da Silva and colleagues aimed at deciphering whether elderly leprosy patients display larger Reactive Oxygen Species (ROS, that cause cell damage) production and its possible impact on the course of the disease. They evaluated the expression of some antioxidant and oxidative burst enzymes in 87 leprosy patients and 25 healthy controls by qPCR in both skin lesions and whole blood, and measured the presence of two oxidative damage markers at several time points (from time at diagnosis until two years after treatment completion). The authors show that elderly patients display increased oxidative damage in skin and whole blood compared to younger patients and controls, and suggest such patients could benefit from antioxidants in addition to the usual treatment. Importantly, the leprosy phenotype of each patient is well defined using the Ridley and Jopling classification. Overall, I have minor comments that should be addressed to increase rigor and clarity.

PLOS authors have the option to publish the peer review history of their article (what does this mean?). If published, this will include your full peer review and any attached files.

Reviewer #1: No

Reviewer #2: No
---

## [Editor Report · Decision Letter 1]

6 Feb 2021

Dear Dr. Esquenazi,

We are pleased to inform you that your manuscript 'Increased oxidative stress in elderly leprosy patients is related to age but not to bacillary load' has been provisionally accepted for publication in PLOS Neglected Tropical Diseases.

Best regards,

Vinicius M Fava, PhD

Associate Editor

Gerd Pluschke

Deputy Editor

---

## [Editor Report · Acceptance letter]

4 Mar 2021

Dear Dr. Esquenazi,

We are delighted to inform you that your manuscript, "Increased oxidative stress in elderly leprosy patients is related to age but not to bacillary load," has been formally accepted for publication in PLOS Neglected Tropical Diseases.

Best regards,

Shaden Kamhawi

co-Editor-in-Chief

Paul Brindley

co-Editor-in-Chief
